# Image-Based Quantitative Analysis of Epidermal Morphology in Wild Potato Leaves

**DOI:** 10.3390/plants13213084

**Published:** 2024-11-01

**Authors:** Ulyana S. Zubairova, Ivan N. Fomin, Kristina A. Koloshina, Alisa I. Barchuk, Tatyana V. Erst, Nadezhda A. Chalaya, Sophia V. Gerasimova, Alexey V. Doroshkov

**Affiliations:** 1The Federal Research Center, Institute of Cytology and Genetics, Siberian Branch of the Russian Academy of Sciences, 630090 Novosibirsk, Russia; infomin82@gmail.com (I.N.F.); kristina.koloshina@yandex.ru (K.A.K.); aliska_chel@mail.ru (A.I.B.); erst@bionet.nsc.ru (T.V.E.); gerson@bionet.nsc.ru (S.V.G.); ad@bionet.nsc.ru (A.V.D.); 2Department of Information Technologies, Novosibirsk State University, 630090 Novosibirsk, Russia; 3N.I. Vavilov Institute of Plant Genetic Resources (VIR), 190000 St. Petersburg, Russia; n.chalaya@vir.nw.ru; 4Department of Genomics and Bioinformatics, Institute of Fundamental Biology and Biotechnology, Siberian Federal University, 660036 Krasnoyarsk, Russia

**Keywords:** *Solanum*, leaf epidermal pattern, pavement cells, stomata, trichomes, plant phenology, plant phenotyping, plant microscopy

## Abstract

The epidermal leaf patterns of plants exhibit remarkable diversity in cell shapes, sizes, and arrangements, driven by environmental interactions that lead to significant adaptive changes even among closely related species. The Solanaceae family, known for its high diversity of adaptive epidermal structures, has traditionally been studied using qualitative phenotypic descriptions. To advance this, we developed a workflow combining multi-scale computer vision, image processing, and data analysis to extract digital descriptors for leaf epidermal cell morphology. Applied to nine wild potato species, this workflow quantified key morphological parameters, identifying descriptors for trichomes, stomata, and pavement cells, and revealing interdependencies among these traits. Principal component analysis (PCA) highlighted two main axes, accounting for 45% and 21% of variance, corresponding to features such as guard cell shape, trichome length, stomatal density, and trichome density. These axes aligned well with the historical and geographical origins of the species, separating southern from Central American species, and forming distinct clusters for monophyletic groups. This workflow thus establishes a quantitative foundation for investigating leaf epidermal cell morphology within phylogenetic and geographic contexts.

## 1. Introduction

The epidermis is the outer cell layer that covers the surface of plants, composed of various cell types [1]. The functions of the epidermis include protecting the plant from harmful environmental factors, maintaining basic physiological processes, and contributing to the mechanical properties of the plant body. The coordinated action of different epidermal cells underlies a wide range of plant physiological properties, which are often linked to agronomically important traits. In the leaf epidermis of most plant species, three easily distinguishable cell types are found: pavement cells, stomata, and trichomes [2]. Pavement cells represent the basic cell type covering the leaf surface, and, while their size and shape vary widely among species, they often resemble jigsaw puzzle pieces [3,4]. The interlocking nature of pavement cells provides the mechanical tension required for proper leaf development and function, and also helps position and organize other cell types [5]. Stomatal guard cells, another specific epidermal cell type, are responsible for regulating gas and water exchange. These cells are always present as symmetric pairs forming the stomatal pore, which they regulate by turgor-driven movements [6]. Trichomes are unicellular or multicellular structures that originate from the epidermis and protrude above its surface. They are generally classified into two types: non-glandular trichomes, or leaf hairs, and glandular trichomes, which produce and secrete specific secondary metabolites [7]. The diversity of trichome structures, functions, and patterns is extremely high among plant species, and the comprehensive classification of trichomes remains an open issue [8].

All three types of epidermal cells can be found on the abaxial (lower) and adaxial (upper) sides of the leaf blade [9]. However, the relative abundance, shapes, and patterns of these cells can vary significantly between the two sides. The mechanisms underlying the formation and diversity of epidermal cell patterns are not well understood [10]. Models for the genetic control of epidermal cell patterning have been established in *Arabidopsis thaliana* for stomata [11], trichomes [12], and pavement cells [13,14], but these models do not consider all cell types simultaneously. Furthermore, in crop species, these *Arabidopsis* models are often not applicable for evaluating the relationship between epidermal cell patterning and agronomically valuable traits. Such evaluations could be highly beneficial for breeding programs, as they provide support for selecting resistant phenotypes and improving epidermis-related traits. The comprehensive investigation and structural modeling of plant epidermal cell patterning across a wide range of species and genotypes require the development of advanced visualization and recognition methods for pattern segmentation, as well as algorithms for differentiating and prioritizing stable and variable traits. Only a few studies have characterized and compared epidermal cell patterns across different plant taxa [15,16], and most existing leaf structure characterization studies focus on closely related groups of species. For example, leaf anatomy and micromorphology have been investigated using light and scanning electron microscopy to understand the correlation of these traits with molecular phylogenetic relationships and environmental adaptations in *Vitis* [17] and *Oxalidaceae* [18]. Additionally, conventional methods have shown great variability in epidermal structures, especially trichomes, among closely related species [19].

Solanaceae crops are of great importance to modern agriculture, with potato being a key species among dicots. Machine-based phenotyping of potato epidermis-related traits could open up new opportunities for breeding programs. Wild relatives of cultivated potato harbor significant genetic diversity [20,21] and serve as valuable sources of traits for potato breeding, including resistance to abiotic stress and pathogens [22]. These traits are often related to plant surface structure and epidermal properties. Interestingly, wild potato species demonstrate enormous diversity in epidermal structures [23], leaves [24], leaf wettability [25], and tubers [26].

This study aims to develop image-based methods for the comprehensive assessment of potato epidermal patterns, including the quantitative evaluation of all epidermal cell types and their interactions. The developed technique was applied to a number of wild potato relatives, and the observed phenomena are discussed in the context of evolution and geographic origin.

## 2. Materials and Methods

### 2.1. Plant Material and Growth Conditions

The following wild potato species from the VIR (Federal Research Center N.I. Vavilov All-Russian Institute of Plant Genetic Resources) genebank collection were used in this experiment: *Solanum demissum*, *S. polyadenium*, *S. stoloniferum*, *S. jamesii*, *S. tarijense*, *S. cardiophyllum*, *S. pinnatisectum*, *S. dolichostigma* (syn. *S. chacoense*), and *S. kurtzianum* (accession numbers are provided in Table 1). The in vitro grown plants were transferred to 15-L pots filled with a natural soil mixed in a 1:1 ratio with Terra Vita universal potting soil (ZAO MNPP FART, St. Petersburg, Russia). The pots were placed outdoors and protected with garden fabric. The plant material for this study was cultivated in 2020–2021 at the same location in the Novosibirsk region (Michurinsky settlement, 54°53′04.1″ N, 82°59′58.6″ E). Eleven weeks after planting, samples of mature leaves from the tips of young plant shoots were collected.

Healthy leaf samples were collected from mature, pot-grown plants during dry weather. For the study of trichome patterns, leaf samples were placed in appropriately sized zip-lock bags to prevent moisture loss. These bags were stored in a refrigerator at 4 °C for a few days (ranging from 1 to 4 days, depending on the imaging schedule), and the bags were opened shortly before imaging. For the study of stomata and pavement cell patterns, leaf blade fragments, with the central vein removed, were fixed in 4% paraformaldehyde.

### 2.2. Image Acquisition of Leaf Epidermis

#### 2.2.1. Trichomes

Following the protocol outlined in [28], leaves were bent in half so that the trichomes lay parallel to the slide at the abaxial side fold area. The leaves were then mounted on a slide using transparent duct tape. For leaves measuring between 1 and 2 cm in length, a single bend was made. For leaves longer than 2 cm, the leaf was cut in half, and a bend was made in each half. Microscopic images were captured using a transmitted light microscope, an Axio Scope A1 (ZEISS, Jena, Germany), equipped with a CCD camera and AxioCam 512 color (ZEISS, Jena, Germany) and controlled by ZEN 2.3 software (ZEISS, Jena, Germany). The imaging setup included an A-Plan 5×/0.15 EC Plan-NEOFLUAR objective lens (ZEISS, Jena, Germany), a Camera Adapter 60-C 1” 1.0× (ZEISS, Jena, Germany), and a total magnification of 50× (10× ocular, 5× objective, 1.0× adaptor).

#### 2.2.2. Pavement Cells and Stomata

Following the protocol described in [29], leaf fragments were stained with DAPI (4′,6-diamidino-2-phenylindole, dihydrochloride, Sigma-Aldrich, St. Louis, MO, USA), Calcofluor White (Sigma-Aldrich, St. Louis, MO, USA), and Propidium Iodide (Sigma-Aldrich, St. Louis, MO, USA). The stained leaf fragments, each with an approximate area of 1.86 mm^2^ and a volume of about 0.09 mm^3^, were scanned using a laser scanning microscope, an LSM 780 NLO (Zeiss, Jena, Germany), in tile scan mode.

### 2.3. Image Analysis and Data Processing

To estimate the distributions of trichome length and density, a computer analysis of photomicrographs from the transverse fold line of the leaf was performed. The process of extracting leaf pubescence characteristics through image analysis is described in detail in [28,30]. This technique provides a fast and accurate method for the quantitative evaluation of trichome length and density.

Reconstruction of the epidermal architecture from laser scanning microscopy images was carried out following the protocol in [29]. Using the Fiji plugin LSM-W^2^, 3D images were converted into 2D surface projections, after which epidermal cells, including stomata, were manually segmented in the 2D images.

The segmented images were processed using the Fiji package according to the following steps: (i) The RGB images were split into three channels: red, green, and blue. (ii) By subtracting the red from the green channel, we obtained an image exclusively showing cell outlines, followed by binarization. (iii) Non-plant cell regions surrounding the plant tissue fragment were removed from the analysis by filling them with black. The images were then inverted and saved in PNG format.

Subsequently, the extracted images were analyzed using the ComponentMeasurement function in Mathematica 12 to assess morphometric descriptors of cell shape [31], including the waviness index (“ConvexCoverage”), which is the ratio of the cell area to the area of its convex hull, as well as actual cell area (“Count”), length (“Length”), width (“Width”), elongation (“Elongation”), circularity index (“Circularity”), rectangularity index (“Rectangularity”), perimeter (“PerimeterLength”), convex hull area (“ConvexCount”), and the perimeter of the convex hull (“ConvexPerimeterLength”).

### 2.4. Phylogenetic Analysis

A phylogenetic tree of wild potato species was obtained from the TimeTree database [32]. The topology was reconstructed using data from [33,34,35].

## 3. Results

More than 14,000 leaf epidermal cells from nine wild potato species were analyzed in this study. Leaf epidermal cells exhibit significant variability in both size and shape. For instance, trichomes can reach several millimeters in length, while stomatal cells are relatively small, typically measuring only a few tens of micrometers. Moreover, trichomes add significant three-dimensional structures to the otherwise flat surfaces of the leaf. To account for all of these features, we developed a comprehensive methodology for leaf sample selection, image acquisition, and processing. This approach enables precise quantitative assessment of a wide range of parameters for various types of epidermal pattern elements, including stomatal cells, trichomes, and standard covering epidermal cells. Figure 1 presents a schematic representation of the methodology, from image acquisition to the analysis of extracted numerical characteristics. A detailed description of each stage can be found in Section 2. The core idea of the proposed method is based on the integration of protocols previously suggested by [28,29]. However, in the present study, we propose a different method for fixing leaf samples, which preserves chlorophyll in the cells, thereby enabling additional pseudocolor values during 3D laser scanning. This improvement provides more accurate visualization of cells, especially glandular trichomes, leading to a more precise phenotypic description of the plant’s leaf epidermis. Our comprehensive study revealed differences in the size and shape of epidermal cells, as well as in leaf epidermis architecture, which emerged as adaptations to ecological niches across different species.

The epidermal pattern was analyzed for plant samples from nine wild potato species, listed in Table 1. These species belong to eight series classified according to [27]. Four of them are diploids from North and Central America (*S. polyadenium*, *S. ehrenbergii*, *S. jamesii*, and *S. pinnatisectum*), three are diploids from South America (*S. chacoense*, *S. kurtizianum*, and *S. tarijense*), and two are polyploid species from North and Central America (*S. demissum* and *S. stoloniferum*), which likely descended from a common ancestor that migrated from South America [27]. The hypothesis of South American origin for these species is supported by molecular phylogenetics and cytogenetic studies [36]. The selected wild potato species exhibit contrasting leaf structures (Figure 2A) and are divided into two distinct clades based on phylogenetic relationships reconstructed using data from the TimeTree database [32]. One clade includes species from North and Central America, while the other comprises species from South America, as well as polyploid migrants.

Using the developed methodology, we conducted a detailed study of the leaf epidermis for the selected nine wild potato species. Figure 2B presents typical examples of cellular patterns, highlighting the differences in the shape, size, and arrangement of epidermal cells between species, as well as between the abaxial and adaxial sides of the leaf blade. Generally, cells on the abaxial side are smaller, and their walls are more undulating. Stomata are present on both sides of the leaf; however, their density is higher on the abaxial side, and their sizes also vary.

Trichomes are generally classified into glandular and non-glandular types, depending on the presence of a glandular head. Within each group, subtypes are distinguished based on the length and cellular composition of the head, stalk, and base of the trichome. The classification of trichomes in plants of the Solanaceae family, based on their morphology, has been well described in studies dating back to the mid-20th century [38,39,40], and includes seven types of trichomes:ITrichomes with a small glandular head, a long multicellular stalk, and a multicellular base;IINon-glandular trichomes with a long multicellular stalk and a multicellular base;IIINon-glandular trichomes with a long multicellular stalk and a unicellular base;IVGlandular trichomes with a small head, a multicellular stalk, and a unicellular base;VNon-glandular trichomes, shorter than types II and III, with a multicellular stalk and a unicellular base;VITrichomes with a four-celled glandular head, a short unicellular stalk, and a unicellular base;VIITrichomes with an eight-celled glandular head, a short unicellular stalk, and a unicellular base.

Channarayappa et al. [37] further refined the classification by introducing subtypes “a” and “b” within type V and subtypes “a”, “b”, and “c” within type VI for species of the genus *Lycopersicon*. Based on this classification, five trichome types were identified across the nine wild potato species examined in our study (Table 1): types II, III, VI-a, VI-c, and VII (Figure 2C and Appendix A). Notably, in accordance with the expanded classification by [37], type VI-c emerged as the most prevalent glandular trichome type among the species analyzed, whereas type VI-a, characterized by a tetralobulate membrane-bound gland at the tip of a unicellular stalk, was exclusively observed in *S. polyadenium*. It is important to highlight that, in certain studies, type VI-a is alternatively referred to as Type A [41,42].

The analysis of the data obtained using a complex of multi-scale methods revealed correlations between the shape parameters of different types of epidermal cells (Figure 3). For all studied species, the parameters differ between the cells of the abaxial and adaxial sides of the leaf, as is clearly demonstrated for *S. pinnatisectum* (Figure 3A,B). Plants continuously optimize growth processes by regulating cell division and expansion, resulting in epidermal cell size being highly dependent on growth conditions, such as light wavelength and intensity [43], or various stress factors [44,45]. On the one hand, epidermal cell size can vary significantly within the same leaf and between different leaves [46], and its distribution reflects the physiological state of the leaf or the entire plant. On the other hand, epidermal cell shape is less influenced by environmental factors and is more closely related to the plant’s genotype [15]. We described the morphology of pavement cells, stomata, and trichomes using parameters such as area, convex hull area, perimeter, convex hull perimeter, length, width, circularity, rectangularity, convex hull coverage, and elongation (Figure 3C,D). In selecting parameters for analysis, we distinguished between dimensional and dimensionless ones, ultimately prioritizing shape over size in our attempt to discriminate between species. We performed a correlation analysis for all estimated epidermal cell shape parameters across all wild potato species studied (Figure 3E and Appendix A). The analysis revealed a relationship between the sizes of trichomes and pavement cells on both the abaxial and adaxial sides of the leaf. Additionally, stomatal cell size on the abaxial side of the leaf strongly correlates with pavement cell size and, interestingly, with glandular trichome length.

An individual analysis of epidermal cell morphology for each studied species (Figure 4A) using principal component analysis (PCA) identified two main factors, accounting for 45% and 21% of the variance, respectively (Figure 4B). These factors correspond to quantitative traits characterizing epidermal cell morphology: the shape of guard cells and the length of non-glandular trichomes in comparison to stomatal and trichome density, as well as the length of glandular trichomes. The species’ positions within this space correlate well with their historical and geographical origins. PCA axes 1 and 2 effectively separated species from southern regions from those found in Central America. Moreover, monophyletic groups of species formed compact, non-overlapping clusters in the space defined by the first two components (Figure 4C). Overall, these results highlight the significant influence of factors related to species’ origins.

## 4. Discussion

Cell morphology is characterized by parameters of size and shape, with epidermal cell size being highly influenced by growth conditions such as light wavelength and intensity [43], as well as various stress factors [44,45]. Plants continuously optimize growth through the regulation of cell division and expansion. The size of epidermal cells can vary significantly within a single leaf blade and between different leaves [46], with this distribution serving as a reflection of the physiological state of the leaf or the entire plant. In contrast, the shape of epidermal cells is more consistent, being less affected by environmental factors and more stable across similar plant organs, such as leaves from the same plant. Thus, epidermal cell shape can be considered as genotype dependent, providing a potential basis for species classification [15].

Despite the wide diversity in leaf epidermal cell shapes, certain trends can be observed across different plant taxa. While pavement cells of closely related species may share specific shape characteristics, it is challenging to establish a comprehensive taxonomic classification based solely on pavement cell morphology across the full diversity of vascular plants [15]. Cell shape diversifies through different mechanisms in various plant taxa, with the molecular mechanisms underlying cell morphology being either conserved or specific to certain clades. On the other hand, the diversification of epidermal cell shapes within narrowly defined taxonomic groups may reflect adaptive microevolutionary processes and could be instrumental for the classification of closely related species.

In addition to pavement cells, trichomes represent a diverse group of epidermal structures that can be classified into distinct types based on their morphology and function. Closely related species often possess a limited number of trichome types, with differences primarily observed in trichome density and type distribution [47]. The taxonomic significance of trichomes remains controversial [47,48]. While attempts to use trichome parameters for plant classification have shown their value at lower taxonomic levels, even at the genus level, trichome diversity is often too high to define specific features [49]. A similar conclusion has been drawn for stomatal micromorphology. Due to the significant variation in structure, shape, and size of stomata, stomatal characteristics have limited taxonomic value [50]. It can be hypothesized that epidermal patterning contributes to the general adaptive strategy of plants, with changes in epidermal patterns being one of the earliest inherited features responding to environmental changes. Given that each type of epidermal cell can be described by its unique quantitative characteristics, a comprehensive assessment of epidermal morphology may reveal specific parameter sets that allow for the discrimination of closely related species and the reconstruction of phylogenies within narrow taxonomic groups.

This study represents the first attempt to develop a comprehensive image-based pipeline for the evaluation of all types of epidermal structures and the identification of parameters that differentiate closely related species. Our approach combines previously developed methods [29,51] that have proven effective for the assessment of trichomes and pavement cells.

In our study, we focused on several accessions from species within the Petota section of the genus *Solanum* to represent closely related species with considerable phenotypic variation. The Petota section is known for its high taxonomic complexity, comprising tuber-forming wild potato species. The section’s diversity, driven by interspecific hybridization, polyploidy, and geographic range, continues to challenge taxonomic classification [36]. Within the species examined here, glandular trichomes were prominent as phenotypic traits useful for taxonomic purposes. Type VI-c was the most widespread glandular trichome type, while type VI-a, characterized by a tetralobulate membrane-bound gland at the tip of a unicellular stalk, was exclusively found in *S. polyadenium*.

Species within the genus *Solanum* exhibit a highly diverse array of trichome morphologies [52]. Although this study does not directly address the molecular biological aspects underlying this morphological diversity, it is essential to recognize that such variation inherently suggests a corresponding diversity in genetic regulation. In the case of trichomes, the fundamental genetic mechanisms underlying trichome patterning are similar to those observed in arabidopsis. As described by Chalvin et al. [53], key initiation genes exhibit a degree of functional conservation. However, Solanum species possess many unique genetic components. Notably, type VI glandular trichomes, which hold significant agricultural value due to their secretory function, are influenced by distinct genetic factors. For instance, the *HAIRS ABSENT* (*H*) gene has been shown to play a critical role in the development of type VI glandular trichomes [54]. Moreover, enzymatic and structural components involved in the biosynthesis of compounds such as terpenoids have been identified, and enhancing these pathways can lead to the expansion of specific trichome cells and the accumulation of these metabolites, rendering them more toxic to herbivorous insects. Vendemiatti et al. [55] provide a detailed, step-by-step account of how resistance to insect pests is mediated by type IV trichomes in tomato. This highlights the potential for breeding programs to develop cultivars with trichomes optimized for both morphology and chemical composition. The introgression of terpenoid biosynthesis genes from wild species has been shown to alter trichome morphology and increase their toxicity to insects [56]. Therefore, understanding the morphology of trichomes provides a foundation for identifying the genetic material necessary for future crop improvement strategies.

Phylogenetically, most wild potato species are divided into two major clades corresponding to the North and Central American region and the South American region [57]. In this study, we examined the epidermal patterns of samples from nine wild potato species belonging to six series as classified by Hawkes et al. [27] (Table 1). Four of these species are diploids originating from North and Central America (*S. polyadenium*, *S. ehrenbergii*, *S. jamesii*, and *S. pinnatisectum*), three are diploids from South America (*S. chacoense*, *S. kurtizianum*, and *S. tarijense*), and two are North and Central American polyploids (*S. demissum* and *S. stoloniferum*), likely derived from a common ancestor that migrated from South America [27]. The hypothesis of a South American origin for these species is supported by both molecular phylogenetic and cytogenetic studies [36]. The reconstruction of phylogenetic relationships among the studied species is depicted in Figure 2. The species are divided into two clades based on their geographic origin; North and Central American species form one clade, while South American species, together with North and Central American polyploids, form a second clade.

## 5. Conclusions

Based on the comprehensive analysis of epidermal cell morphology in nine wild potato species, significant variations in cell size, shape, and arrangement were identified across different species and between the abaxial and adaxial sides of the leaf. These variations are strongly influenced by both genetic and environmental factors, with cell shape being more genotype dependent, while cell size is more responsive to external conditions such as light and stress. The principal component analysis (PCA) revealed distinct morphological clusters that correlate with the geographical and evolutionary origins of the species, effectively separating species from southern regions and Central America. The correlation analysis further highlighted important relationships between trichome size, stomata density, and the morphology of pavement cells, underscoring the role of these features in adaptation to diverse ecological niches. Overall, these findings demonstrate the utility of combining multi-scale methods to characterize epidermal patterns and provide insights into the adaptive significance of epidermal traits in wild potato species.

## Figures and Tables

**Figure 1 plants-13-03084-f001:**
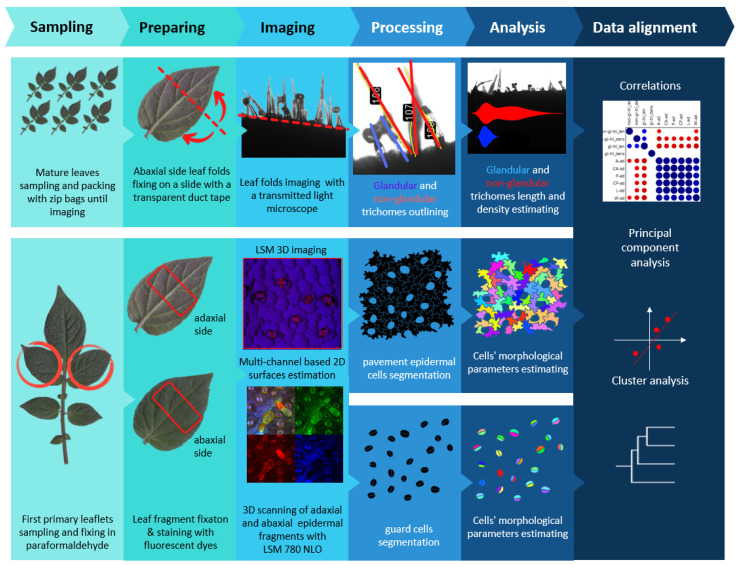
A workflow of microscopy and computer vision methods for quantifying morphological parameters of leaf epidermal patterns. The stages of the experimental protocol, along with the intermediate steps of data acquisition, analysis, and integration, are shown on colored backings.

**Figure 2 plants-13-03084-f002:**
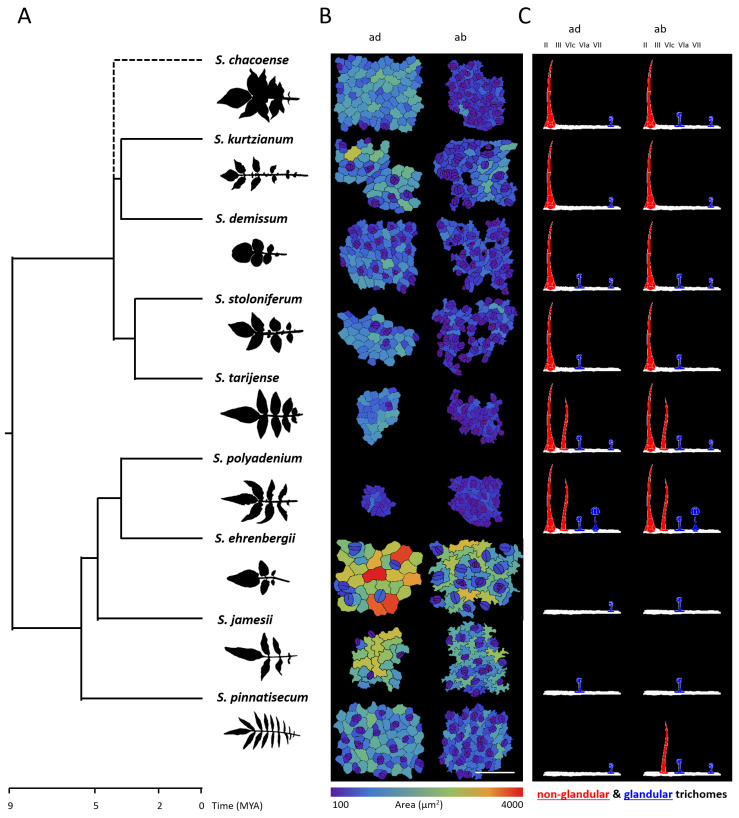
Morphological features of the epidermal leaf pattern in nine wild potato species. (**A**) Phylogenetic relationships among the species and the shape of their leaves. (**B**) The morphology of pavement cells and stomata in the leaf epidermis. (**C**) Trichome types specific to each species. “ab” and “ad” refer to the abaxial and adaxial sides of the leaf, respectively. Types II, III, VI-c, VI-a, and VII are referred to according to classification from [37].

**Figure 3 plants-13-03084-f003:**
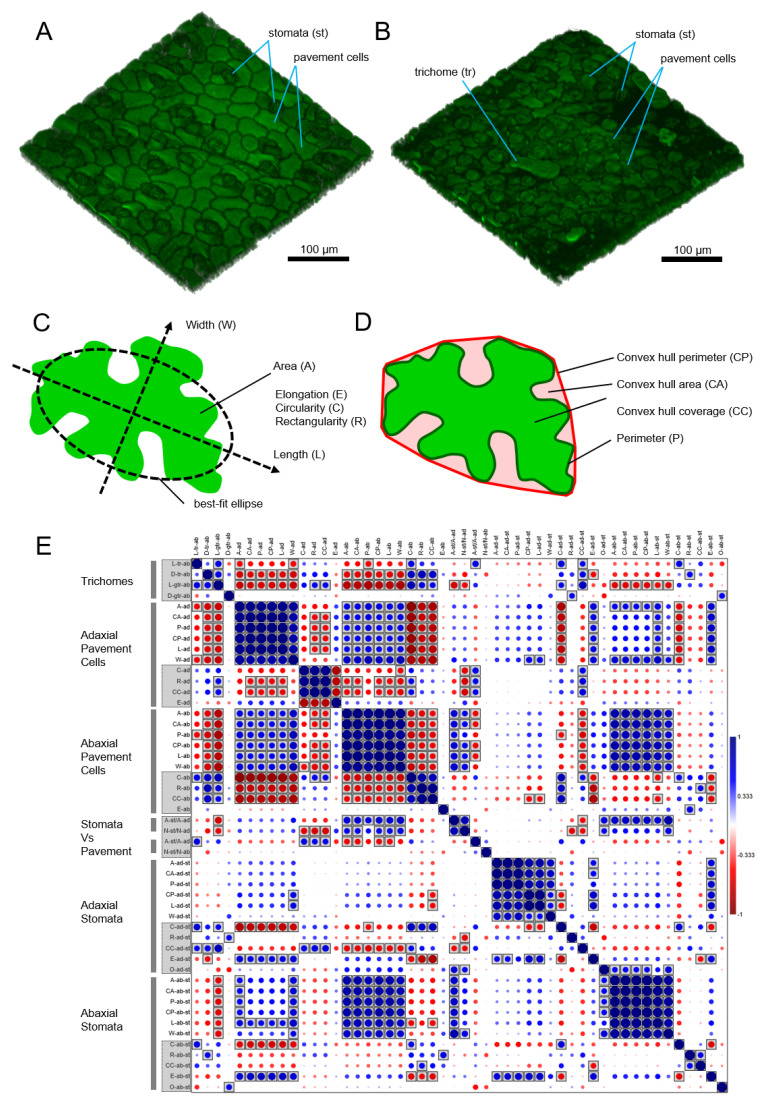
Dimensional and dimensionless parameters of leaf epidermal cell morphology in wild potatoes. Samples of 3D images of adaxial (**A**) and abaxial (**B**) leaf epidermis of *S. pinnatisectum*, showing different epidermal cell types (pavement cells, stomata, trichomes). (**C**) Evaluation of dimensional parameters and (**D**) dimensionless parameters of epidermal cell shape. (**E**) Correlation matrix (Kendall tau) for the estimated epidermal cell shape parameters in the wild potato species studied. Parameters are grouped according to cell types and are highlighted with gray lines for further analysis. Abbreviations for parameters: for morphometric parameters, L is length, D is density, A is area, CA is convex hull area, P is perimeter, CP is convex hull perimeter, W is width, C is circularity, R is rectangularity, CC is convex hull coverage, E is elongation; for cell types, tr is trichome, gtr is glandular trichome, st is stomata, and there is no designation for pavement cells; ab is the abaxial and ad is the adaxial side of the leaf.

**Figure 4 plants-13-03084-f004:**
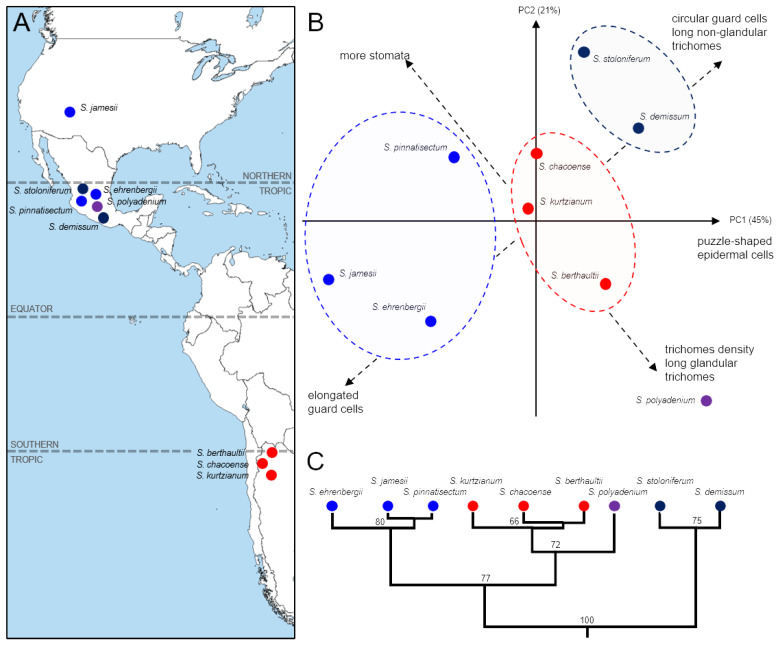
Comparison between geographical origins of wild potato species and morphological characteristics of leaf epidermal cells: (**A**) Proposed geographical origins of wild potato species. (**B**) Principal component analysis (PCA) and (**C**) cluster analysis of nine wild potato species, based on pairwise distances between the medians of morphological characteristics of the leaf epidermis.

**Table 1 plants-13-03084-t001:** Wild potato species utilized for epidermal pattern analysis.

Species Name ^a^	Accession No. VIR Collection	Accession No. Other Databases	Ploidy	EBN
*S. demissum* Lindl.	15,176	PI 275211	6n	4
*S. polyadenium* Greenm.	24,957	PI 230480	2n	2
*S. stoloniferum* Schltdl.	3326	N/A	4n	2
*S. jamesii* Torr.	24,923	PI 612450	2n	1
*S. tarijense* Hawkes	12,637	PI 473217	2n	2
*S. cardiophyllum* Lindley	16,828	PI 283063	2n	1
*S. pinnatisectum* Bitter	23,569	PI 253214	2n	1
*S. dolichostigma* Buk. (syn. *S. chacoense* Bitter)	7613	N2892	2n	2
*S. kurtzianum* Bitter and Wittm	11,969	N/A	2n	2

^a^ Taxonomy based on [27].

## Data Availability

All quantitative data generated and analyzed in this study can be found in Appendix A. Microscopy images reported in this paper will be shared by the lead contact upon request.

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
