# Peer review of "Image-Based Quantitative Analysis of Epidermal Morphology in Wild Potato Leaves"

_plants, 2024, doi:10.3390/plants13213084_

Round 1

Reviewer 1 Report

Comments and Suggestions for Authors

Epidermal structures, especially trichomes, have an important impact on plant-ppredator interaction. Unfortunately, many cultivars of economic plants are lacking trichomes qualitatively or quantitatively, making them more accessible for predators. The search for genetic resources in closely related species is therefore important for future resistance breeding.

The paper is part of a series dealing with potato epidermal structures. The methods are very fine and show here a process flow that enables a semi-automatic phenotypic throughput.

The paper is well done and deserves to be published after commenting or improving my only concern: on page 7, the authors describe the trichome classification performed on Lycopersicon (Ref [37] Channarayappa et al., 1992). Unfortunately, I do not have access to this paper to clarify details. 

The classification mentions 7 types and subtypes in types V and VI. Type VI has three substpyes (a, b, and c). The authors describe in their paper, relying on the classification of ref37 a subtype VId. Is this a new type?

The authors describe in discussion the taxonomic implications of trichome types as 'controversial', presenting good references pro and contra. But the contribution of their work to this discussion is not enough emphasized. 

The classification of such trichome types should, in my experience, not be as narrow because of the variability of some morphological features leading to false classifications. If the authors go up a level and simplify the classification by uniting some close types, is there more information visible?

Author Response

Dear Reviewer,

Thank you very much for your positive assessment of our work and for your comments.

Comments 1: on page 7, the authors describe the trichome classification performed on Lycopersicon (Ref [37] Channarayappa et al., 1992). Unfortunately, I do not have access to this paper to clarify details. The classification mentions 7 types and subtypes in types V and VI. Type VI has three substpyes (a, b, and c). The authors describe in their paper, relying on the classification of ref37 a subtype VId. Is this a new type?

Response 1: We acknowledge that during the preparation of the manuscript, we mistakenly confused "a" and "d" due to their similar spelling. We appreciate your careful review and for bringing this error to our attention. The necessary corrections have been made to the text, Figure 2, and Supplementary table 1 to accurately reflect the intended information. Thank you again for your valuable feedback. We are attaching a pdf of the article Channarayappa et al., 1992 so that you can read it.

Original text in the manuscript (Lines 190-196, page 7):

Additionally, the study by [37] introduced subtypes "a" and "b" within type V, as well as subtypes "a", "b", and "c" within type VI for species of the genus Lycopersicon. According to this classification, five types of trichomes were identified in the wild potato species studied (Table 1): Types II, III, VI-c, VI-a, and VII (Figure 2C and Supplementary table 1). Interestingly, according to the extended classification from [37], type VI-c was found to be the most widespread glandular trichome type among the species we studied. Type VI-d was found only in S. polyadenium.

New text:

Channarayappa et al. [37] further refined the classification by introducing subtypes "a" and "b" within type V, and subtypes "a", "b", and "c" within type VI for species of the genus Lycopersicon. Based on this classification, five trichome types were identified across the nine wild potato species examined in our study (Table 1): types II, III, VI-a, VI-c, and VII (Figure 2C and Supplementary Table 1). Notably, in accordance with the expanded classification by [37], type VI-c emerged as the most prevalent glandular trichome type among the species analyzed, whereas type VI-a, characterized by a tetralobulate membrane-bound gland at the tip of a unicellular stalk, was exclusively observed in S. polyadenium. It is important to highlight that in certain studies, type VI-a is alternatively referred to as Type A [ave1986phenolic, gibson1971glandular]

New refs:

Avé, D. A., & Tingey, W. M. (1986). Phenolic constituents of glandular trichomes on Solanum berthaultii and S. polyadenium. American potato journal, 63, 473-480.

Gibson, R. W. (1971). Glandular hairs providing resistance to aphids in certain wild potato species. Annals of Applied Biology, 68(2), 113-119.

Comments 2: The authors describe in discussion the taxonomic implications of trichome types as 'controversial', presenting good references pro and contra. But the contribution of their work to this discussion is not enough emphasized. The classification of such trichome types should, in my experience, not be as narrow because of the variability of some morphological features leading to false classifications. If the authors go up a level and simplify the classification by uniting some close types, is there more information visible?

Response 2: Thank you very much, we took your comment into account and added the corresponding text to the Discussion section.

Original text in the manuscript (Lines 273-279):

For this study, we selected a few accessions from different species within the Petota section of the genus Solanum as examples of closely related but phenotypically diverse species. The Petota section is a highly diverse taxonomic group that includes tuber-bearing wild potato species. Due to the potential for interspecific hybridization, various forms of polyploidy, and broad geographic distribution associated with phenotypic plasticity, the taxonomy of wild potatoes remains an open question [52].

New text:

In our study, we focused on several accessions from species within the Petota section of the genus Solanum to represent closely related species with considerable phenotypic variation. The Petota section is known for its high taxonomic complexity, comprising tuber-forming wild potato species. The section’s diversity, driven by interspecific hybridization, polyploidy, and geographic range, continues to challenge taxonomic classification \citep{spooner2014systematics}. Within the species examined here, glandular trichomes were prominent as phenotypic traits useful for taxonomic purposes. Type VI-c was the most widespread glandular trichome type, while type VI-a, characterized by a tetralobulate membrane-bound gland at the tip of a unicellular stalk, was exclusively found in S. polyadenium.

Reviewer 2 Report

Comments and Suggestions for Authors

In this manuscript, the authors investigated the morphological parameters of potato leaves based on a self-made workflow. The results are interesting and helpful for deepening our understanding of the taxonomy and evolution of wild potato species. Most of the data are comparable to related publications, and new taxonomical approaches were provided based on digital image analysis. This is inspiring for designing new methods to promote traditional taxonomy. The results are clear and well organized.

Minor points

I suggest the authors compare the morphological results with published molecular biological data more carefully. In the discussion, lines 278-289, the authors mentioned several publications. Nevertheless, I believe the identity and difference between results from designed morphological methods, traditional morphological methods, and molecular biological methods should be compared and discussed more carefully.   

Author Response

Dear Reviewer,

Thank you very much for your positive assessment of our work and for your comments.

Comments 1: I suggest the authors compare the morphological results with published molecular biological data more carefully. In the discussion, lines 278-289, the authors mentioned several publications. Nevertheless, I believe the identity and difference between results from designed morphological methods, traditional morphological methods, and molecular biological methods should be compared and discussed more carefully.

Response 1: In order to correlate our morphological findings with previously published molecular biological data, we have incorporated the following paragraph into the Discussion section (before Lines 278-289):

New text:

Species within the genus Solanum exhibit a highly diverse array of trichome morphologies \cite{watts2021morphological}. Although this study does not directly address the molecular biological aspects underlying this morphological diversity, it is essential to recognize that such variation inherently suggests a corresponding diversity in genetic regulation. In the case of trichomes, the fundamental genetic mechanisms underlying trichome patterning are similar to those observed in Arabidopsis. As described by Chalvin et al. \cite{chalvin2020genetic}, key initiation genes exhibit a degree of functional conservation. However, Solanum species possess many unique genetic components. Notably, type VI glandular trichomes, which hold significant agricultural value due to their secretory function, are influenced by distinct genetic factors. For instance, the HAIRS ABSENT (H) gene has been shown to play a critical role in the development of type VI glandular trichomes \cite{gasparini2023natural}. Moreover, enzymatic and structural components involved in the biosynthesis of compounds such as terpenoids have been identified, and enhancing these pathways can lead to the expansion of specific trichome cells and the accumulation of these metabolites, rendering them more toxic to herbivorous insects. Vendemiatti et al. \cite{vendemiatti2022genetic} provide a detailed, step-by-step account of how resistance to insect pests is mediated by type IV trichomes in tomato. This highlights the potential for breeding programs to develop cultivars with trichomes optimized for both morphology and chemical composition. The introgression of terpenoid biosynthesis genes from wild species has been shown to alter trichome morphology and increase their toxicity to insects \cite{therezan2021introgression}. Therefore, understanding the morphology of trichomes provides a foundation for identifying the genetic material necessary for future crop improvement strategies.

New refs:

Chalvin, C., Drevensek, S., Dron, M., Bendahmane, A., & Boualem, A. (2020). Genetic control of glandular trichome development. Trends in Plant Science, 25(5), 477-487.

Therezan, R., Kortbeek, R., Vendemiatti, E., Legarrea, S., de Alencar, S. M., Schuurink, R. C., ... & Peres, L. E. (2021). Introgression of the sesquiterpene biosynthesis from Solanum habrochaites to cultivated tomato offers insights into trichome morphology and arthropod resistance. Planta, 254(1), 11.

Gasparini, K., Gasparini, J., Therezan, R., Vicente, M. H., Sakamoto, T., Figueira, A., ... & Peres, L. E. (2023). Natural genetic variation in the HAIRS ABSENT (H) gene increases type-VI glandular trichomes in both wild and domesticated tomatoes. Journal of Plant Physiology, 280, 153859.

Watts, S., & Kariyat, R. (2021). Morphological characterization of trichomes shows enormous variation in shape, density and dimensions across the leaves of 14 Solanum species. AoB Plants, 13(6), plab071.

Vendemiatti, E., Therezan, R., Vicente, M. H., Pinto, M. D. S., Bergau, N., Yang, L., ... & Peres, L. E. (2022). The genetic complexity of type-IV trichome development reveals the steps towards an insect-resistant tomato. Plants, 11(10), 1309.